# Thermoresponsive Bentonite for Water-Based Drilling Fluids

**DOI:** 10.3390/ma12132115

**Published:** 2019-06-30

**Authors:** Wenxin Dong, Xiaolin Pu, Yanjun Ren, Yufen Zhai, Feng Gao, Wei Xie

**Affiliations:** 1State Key Lab of Oil and Gas Reservoir Geology and Exploitation, Southwest Petroleum University, Chengdu 610500, China; 2College of Chemistry and Chemical Engineering, Southwest Petroleum University, Chengdu 610500, China

**Keywords:** bentonite, N-isopropylacrylamide, drilling fluid, self-recovery

## Abstract

As an important industrial material, bentonite has been widely applied in water-based drilling fluids to create mud cakes to protect boreholes. However, the common mud cake is porous, and it is difficult to reduce the filtration of a drilling fluid at high temperature. Therefore, this paper endowed bentonite with a thermo response via the insertion of N-isopropylacrylamide (NIPAM) monomers. The interaction between NIPAM monomers and bentonite was investigated via Fourier infrared spectroscopy (FTIR), isothermal adsorption, and X-ray diffraction (XRD) at various temperatures. The results demonstrate that chemical adsorption is involved in the adsorption process of NIPAM monomers on bentonite, and the adsorption of NIPAM monomers accords with the D–R model. With increasing temperature, more adsorption water was squeezed out of the composite when the temperature of the composite exceeded 70 °C. Based on the composite of NIPAM and bentonite, a mud cake was prepared using low-viscosity polyanionic cellulose (Lv-PAC) and initiator potassium peroxydisulfate (KPS). The change in the plugging of the mud cake was investigated via environmental scanning electron microscopy (ESEM), contact angle testing, filtration experiments, and linear expansion of the shale at various temperatures. In the plugging of the mud cake, a self-recovery behavior was observed with increasing temperature, and resistance was observed at 110 °C. The rheology of the drilling fluid was stable in the alterative temperature zone (70–110 °C). Based on the high resistance of the basic drilling fluid, a high-density drilling fluid (ρ = 2.0 g/cm^3^) was prepared with weighting materials with the objective of drilling high-temperature formations. By using a high-density drilling fluid, the hydration expansion of shale was reduced by half at 110 °C in comparison with common bentonite drilling fluid. In addition, the rheology of the high-density drilling fluid tended to be stable, and a self-recovery behavior was observed.

## 1. Introduction

As an important industrial material, bentonite has been widely applied in agricultural, medical, and energy fields for decolorization, suspension, and stabilization [1,2,3]. During drilling for oil or gas, bentonite can effectively reduce the loss of the filtration of the water-based drilling fluid and improve the ability of the water-based drilling fluid to shear and carry cuttings [4,5]. Under deep stratum pressure, the water-based drilling fluid can quickly lose a substantial amount of water and form a thin film (mud cake), which consists mainly of bentonite, on the borehole [6]. The film can prevent the residual water of the drilling fluid from hydrating the borehole and collapsing the stratum. However, the simple film cannot effectively prevent the drilling fluid from invading the stratum with the increase of the temperature and of the drilling depth, which typically causes a series of serious problems, such as fracture expansion, borehole collapse, and oil leakage [7,8]. Therefore, it is necessary to further reduce the loss of water from the drilling fluid. Drilling engineers typically utilize a series of high-molecular polymers, such as polyanionic cellulose (PAC), xanthan gum (XG), and carboxymethylcellulose (CMC) to bind water molecules in drilling fluids [9,10,11]. However, this can significantly increase the difficulty and cost of drilling with the increase of the viscosity of the fluid and can even cause drill bit sticking and borehole collapse. In contrast, in this paper, we attempt to reduce the water loss by modifying bentonite and tuning the response of the mud cake to temperature. Via this approach, we can design the responsive temperature range of the mud cake to realize the plugging of water.

Huang synthesized a series of P(NIPAM-*co*-AA)/clay nanocomposite hydrogels via radical copolymerization, using clay as the cross-linker. In the research, the composite hydrogels have excellent mechanical strength for a wide range of clay concentrations. However, the thermoresponse of the composite hydrogel was not reported [12]. In addition, a novel octadecyl amine–(ODA–Mt) copolymer nanocomposite was synthesized for bioengineering by modifying the montmorillonite with N-isopropylacrylamide (NIPAM) polymers [13]. In this research, the heating method is demonstrated to be a facile and highly effective technique for clay polymer nanocomposites (CPNs) that provides a higher rate of copolymerization and conversion of micro- and nanoparticles [14]. A clay particle is in the core and thermoresponsive NIPAM polymers are on the shell, as illustrated in Figure 1. The linear NIPAM polymers can cross-link with each other as the temperature increases; and return to their linear conformations as the temperature decreases. Similarly, poly(N-isopropylacrylamide) (PNIPAM)/clay lithium magnesium silicate hydrate (LMSH) nanocomposite hydrogels with various clay percentages of LMSH/NIPAM (which are referred to as NPX hydrogels) were prepared to remove anionic dye Amaranth from the aqueous solution [15]. Although nanocomposites of NIPAM polymers and clay have been applied in a wide range of fields, reports on the application of water-based drilling fluid are rare.

Therefore, in this paper, we attempt to modify bentonite with mono-NIPAM to decrease the water loss of drilling fluids at high temperature while maintaining the rheology of the fluids. Firstly, we designed NIPAM/bentonite composite by mixing NIPAM monomers and bentonite at different temperatures. The interaction between NIPAM monomers and bentonite was investigated via Fourier transform infrared spectroscopy (FTIR) and X-ray diffraction (XRD) at designed temperatures. The composite showed a certain hydrophobicity at 70 °C and a resistance to high temperature (90 °C). Furthermore, we conducted the isothermal adsorption to explore the nature of adsorption. We found the chemisorption was involved, which may be due to the hydrogen bond and the protonation of amine groups of NIPAM monomers. Furthermore, we applied the composite in a potassium peroxydisulfate (KPS) aqueous solution for drilling oil and gas at high temperature. With this excitation, the switching behavior of the composite was extremely stable in water when the serving temperature was above low critical phase transition temperature (LCST). Therefore, the smart water-based drilling fluid can be prepared with NIPAM/bentonite instead of conventional bentonite for drilling oil and gas in deep stratum. The self-recovery behavior was found in the filtration, plugging, and rheology of modified drilling fluid, which showed great potential for exploring shale oil/gas at high temperature.

## 2. Experiment

### 2.1. Material 

Na^+^ bentonite was purchased from the Nanocor Company (Xie et al., 2017). The chemical composition of the bentonite was 13.22% Al_2_O_3_, 71.30% SiO_2_, 7.10% MgO, 4.79% Na_2_O, and 3.59% Fe_2_O_3_ [16]. The cation exchange capacity (CEC) of the bentonite was 145 meq/100 g [17]. The samples were dried at 150 °C and sifted through 200-mesh sieves. The smart N-Isopropylacrylamide (NIPAM) monomer (Sigma-Aldrich, Berlin, Germany) was purified prior to use via recrystallization from diethyl ether. The ratio of amide groups to sec-propyl groups is 1:1, which provided the potential for thermoresponse. Low-viscosity polyanionic cellulose (Lv-PAC) (Aladdin, Shanghai, China) served as a common cross-linker for the drilling fluid [18]. Carboxymethylcellulose (CMC), potassium peroxydisulfate (KPS), barite (BaSO_4_) (with a particle size of 1 μm), and calcium carbonate (CaCO_3_) (with a particle size of 7.5 μm) were also obtained from Aladdin and used to prepare the drilling fluid. The shale was drilled from the Longmaxi formation in Sichuan province of China and used to evaluate the effect of the prepared drilling fluid on the hydration of shale.

### 2.2. Preparation of the NIPAM/Bentonite Composite

The main clay mineral of the bentonite was Na^+^ montmorillonite, the edges of which contain polar hydroxyl groups [19]. Based on this, the composite of NIPAM and bentonite can be simply prepared by mixing them with each other. First, the bentonite was dried at 150 °C for 24 h. Then, 4.0 g bentonite was added into 100 mL NIPAM aqueous solution (the mass percent of NIPAM was 3%) and stirred at 30 °C for 24 h. After that, the mixture was dehydrated at 6000 rpm for 5 min and the residual wet bentonite was collected and washed three times with distilled water. Finally, the dehydrated clay was dried at 80 °C for 24 h and grinded to micrometer scale (200-mesh) with a powder extractor.

### 2.3. Isothermal Adsorption of NIPAM on Bentonite

A total of 0.1 g bentonite (at 150 °C for 24 h) was added into 10 mL NIPAM solution for adsorption at three temperatures (30 °C, 70 °C, and 90 °C). After 24 h, the equilibrium concentration of the residual solution can be obtained from the reading absorbance [20] of a UV-Vis spectrophotometer from the Shanghai Onlab Instruments company (Shanghai, China). In detail, 3.5 mL residual solution was extracted into a preheated cuvette and the absorbance was recorded by the UV-Vis spectrophotometer with scanning wavelength between 320~400 nm [14].

The equilibrium adsorption capacity of bentonite was calculated from the N element content, which was obtained via elemental analysis (EA) using the Var10EL-III analyzer (Elementar, Levokusen, Germany). The initial concentration of the NIPAM solution was controlled from 5 to 45 mg/L.

### 2.4. Structural Characterization

The changes in the basal spacing of the clay interlayer with the function of the NIPAM monomers were recorded via X-ray diffraction (XRD). The XRD patterns were recorded by an X’Pert PRO MPD diffractometer from PANalytical B.V. (Amsterdam, Holland, the Netherlands), which was equipped with a Cu Kα radiation source. The microstructures of hydrated NIPAM/bentonite composites were observed via Quanta 450 environmental scanning electron microscopy (ESEM, FEI, Hillsboro, OR, USA). The interaction between NIPAM monomers and bentonite was identified via Fourier transform infrared spectroscopy (Thermo Scientific, Boston, MA, USA).

### 2.5. Thermal Sensitivity Analysis

The NIPAM/bentonite solution was firstly prepared by stirring NIPAM/bentonite with KPS aqueous solution (the mass percent of KPS was 0.025%) at different temperatures for 30 min. After that, 2 mL active NIPAM/bentonite solution was put into a dried cuvette for investigating the changing of transmittance via UV-1750 Ultraviolet visible spectrophotometer (Shimadzu, Tokyo, Japan). Besides, 30 mL active NIPAM/bentonite solution was poured into the stirred vessel of particle size analyzer (BT-9300LD, Betttersize, Shanghai, China) for investigating the changing of particle size of NIPAM/bentonite.

### 2.6. Preparation of Fresh Mud Cakes from the NIPAM/BENTONITE Composite

Fourteen grams of NIPAM/bentonite composite was prepared and stirred with 350 mL Lv-PAC solution for 16 h. The influence of the temperature on the composite was investigated with a high temperature and high pressure (HTHP) filtration tester (Tongchun, Qingdao, China). First, a filter paper (with a maximum diameter of less than 20 μm) was inserted into the bottom of the tank of the tester. Then, 350 mL prepared slurry was poured into the tank and filtrated at 3.5 MPa at eight temperatures (30, 50, 70, 90, 110, 130, 150, and 180 °C) for 30 min. Next, the residual slurry was poured and the fresh mud cake in the bottom was collected for ESEM analysis and hydrophilic–hydrophobic analysis. 

### 2.7. Conversion of Mud Cakes with the Hydrophilic–Hydrophobic Property

The conversion of mud cakes with the hydrophilic-hydrophobic property was investigated by measuring changes in the contact angle. Fresh mud cakes had been previously prepared at various temperatures and 0.2 mL distilled water was injected on the surface of each mud cake. The contact angle was recorded by a PDE 1700LL/DSA100 HTHP interfacial tension meter (Kruss, Berlin, Germany).

### 2.8. Change in the Filtration of the NIPAM/Bentonite Drilling Fluid with Temperature

The filtration of slurry was monitored using the HTHP filtration tester (Tongchun, China). Into the tank, 350 mL slurry was poured and it was filtrated at 3.5 MPa at eight temperatures (30, 50, 70, 90, 110, 130, 150, and 180 °C) for 1 h. The formulations of the water-based drilling fluids are listed in Table 1.

### 2.9. Change in the Rheology of the NIPAM/Bentonite Drilling Fluid with the Temperature

Into the tank, 300 mL drilling fluid was added and it was rolled at various temperatures for 24 h. Then, the rheology of the aging drilling fluid was measured using a Fann 35A viscosimeter (Tongchun, China). The rheological parameters were calculated from readings in the range of 3 to 600 rpm via the following formulas [17].
*μ_a_* = *ϴ_600_*/2 (mPa·s),(1)
*μ_p_* = *ϴ_600_* − *ϴ_300_* (mPa·s),(2)
*τ_o_* = (*ϴ_300_* − *μ_p_*)/2 (N/m^2^),(3)
where *μ_a_* is the apparent viscosity, *μ_p_* is the plastic viscosity, *τ_o_* is the yield point, and *ϴ_300_* and *ϴ_600_* represent readings at 300 and 600 rpm, respectively.

### 2.10. Inhibition by the NIPAM/Bentonite Drilling Fluid of the Expansion of Shale as a Function of the Temperature

First, the shale was ground and screened using 200-mesh sieves. Next, 10 g shale powder was added into a mold and pressured at 10 MPa for 5 min. After that, the compacted shale was placed in a HTHP dilatometer (Tongchun, China) and 10 mL drilling fluid was injected by N_2_ gas. The linear expansion of the core over 16 h was recorded by the dilatometer. The linear expansion ratio was calculated via the following equation:(4)$$\omega =\left(R_{t}-R_{o}\right)/H\times 100\%,$$
where *ω* is the linear expansion ratio, %; *R_t_* is the reading height at time *t*, mm; *R_o_* is the initial reading, mm; and *H* is the original depth of the shale, mm.

## 3. Discussion and Results

### 3.1. Chemical Characterization of Adsorbed NIPAM Monomers in Bentonite at Various Temperatures

The chemical composition of the NIPAM/bentonite composite was identified via FTIR analysis, as shown in Figure 2. The peaks at 661 cm^−1^ can be assigned to the wagging vibration of water molecules [21], which indicated the adsorption water in bentonite. The peaks at 783 cm^−1^ and 1041 cm^−1^ represent the symmetric and asymmetrical stretching vibrations of the Si–O–Si of montmorillonite [22], respectively. 

Besides, the stretching vibration peak at 1660 cm^−1^ corresponds to the C=O bond of the carbonyl group of the acylamide [23]. The peak at 1624 cm^−1^ corresponds to the conjugative stretching vibration of the C=C of the alkyl group [24]. The deformation vibration absorption peak at 1545 cm^−1^ corresponds to the N–H bond of the secondary amide [25]. The characteristic absorption band peaked at 2958 cm^−1^, which corresponds to the stretching vibration of the C–H bond of the alkyl group [26]. In addition, the sharp peak at 3680 cm^−1^ corresponds to the vibration of the OH groups of bentonite [27]. These peaks show that the composite of NIPAM and bentonite was prepared well.

With increasing temperature, the adsorbed NIPAM monomers cannot be evaporated unless the experimental temperature exceeds 90 °C according to curves K_N_(50) through K_N_(90). Besides, the sharp peak of the OH groups of bentonite disappeared in the composite according to curves K_N_(50) and K_N_(90). Hence, NIPAM monomers may react with OH groups or adsorb with active OH groups with hydrogen bond [28].

### 3.2. Isothermal Adsorption Model of NIPAM Monomers on Bentonite

The adsorption isotherm is an indispensable tool for studying the adsorption mechanism of an adsorption process and can indicate, for example, the adsorption nature of the adsorption process and the adsorption capacity of the adsorbent. The most suitable isotherm model of an adsorption system according to adsorption isotherm and the corresponding model parameters can reveal important information on the adsorption mechanism [15,29].

In Figure 3, the S-shape of the curves indicated three stages of adsorption. At the beginning of S-shape curves the adsorption amount increased smoothly, indicating that NIPAM monomers were simply adsorbed with physical adsorption [30]. In this stage, most of NIPAM monomers may be locked into the micropores of bentonite with capillary force and Van der Waals’ force [31,32]. With the increase of NIPAM concentration, more NIPAM monomers can react with bentonite and the adsorption amount showed an exponential growth, which may involve chemisorption. When the concentration was above 40 mg/L, the adsorption amount tended to the maximum, which indicates adsorption equilibrium.

Besides, the adsorption capacities of NIPAM monomers at several temperatures (30 °C, 70 °C, and 90 °C) are plotted. At the low temperature (30 °C), the NIPAM adsorption increased sharply from 0 to 2.76 mg/g as the NIPAM solution concentration increased from 0 to 20 mg/L. However, the adsorption of NIPAM monomers was obviously decreased when the temperature exceeded the phase transition temperature of the NIPAM solution [33]. Experimentally, the equilibrium adsorption increased slowly from 0 to 2.40 mg/g as the original NIPAM solution concentration increased from 0 to 45 mg/L at high temperature (70 °C). However, most of the NIPAM monomers (q_m_ = 1.79 mg/g) were still adsorbed onto bentonite at 90 °C, even if the NIPAM monomers tended to evaporate in the FTIR analysis. 

To identify the adsorption mechanism of NIPAM, the experimental equilibrium data were fitted to the Langmuir [34], Freundlich [35], Temkin [36], and Dubinin–Raduskevich [30] isotherm equations. The models are described by the following formulas:(5)$$\mathrm{Langmuir}\;\mathrm{model}:\;\frac{C_{e}}{q_{e}}=\frac{C_{e}}{q_{m}}+\frac{1}{q_{m}b},$$
(6)$$\mathrm{Freundlich}\;\mathrm{model}:\;\ln q_{e}=\ln K_{F}+\frac{1}{n}\ln C_{e},$$
(7)$$\mathrm{Temkin}\;\mathrm{model}:\;q_{e}=\mathrm{BlnA}+\mathrm{Bln}C_{e},$$
(8)$$D–R\;\mathrm{model}:\;\ln q_{e}=\ln q_{m}-B_{1}\Sigma^{2},$$
(9)$$\Sigma =RT\ln(1+1/C_{e}),$$
(10)$$E=\frac{1}{\sqrt{2B_{1}}},$$
where *C*_e_ is the equilibrium NIPAM concentration in the solution (mg/L); b is the Langmuir adsorption constant (L/mg); *q_e_* is the equilibrium adsorption capacity of the sample (mg/g); *q_m_* is the theoretical maximum adsorption capacity of the sample (mg/g); K_F_ (L/mg) and n are Freundlich isotherm constants that, respectively, correspond to the capacity for and intensity of the adsorption; A is the equilibrium binding constant (L/mg) and B is related to the heat of adsorption; B_1_ is the D–R model constant (mol^2^kJ^−2^), which is related to the mean free energy of the adsorption per mole of the adsorbate; Σ is the Polanyi potential; and E is the mean free energy of adsorption (kJ/mol). The values of the isotherm parameters are listed in Table 2, according to which the D–R model more accurately describes the adsorption equilibrium of bentonite on NIPAM monomers because it has the highest correlation coefficient values (R^2^, 0.9988, 0.9977, 0.9929) at the experimental temperatures (30 °C, 70 °C, and 90 °C) among the examined models. 

Based on the D–R model, the surface of the composite was not homogeneous, and the adsorption potential was not constant [30]. Using the D–R model, the adsorption type (physical or chemical adsorption) can be determined according to the free energy (E). If the mean free energy (E) is <8 kJ·mol^−1^, physical forces such as Van der Waals forces and hydrogen bonds may affect the adsorption mechanism. The adsorption process is triggered by ion exchange when the value of the mean free energy changes from 8 to 16 kJ·mol^−1^. If the calculated E value is >16 kJ·mol^−1^, the adsorption process is of a chemical nature [20]. According to Table 2, the values of E (17.62, 17.13, 16.82 kJ·mol^−1^) were >16 kJ·mol^−1^, hence, chemical adsorption is involved in the adsorption process of NIPAM monomers onto bentonite, which accords with the conclusion of the FTIR analysis. There may be hydrogen bond and cation exchange between NIPAM monomers and montmorillonite. The active primary amines could generate hydrogen bonds with the hydroxyl groups of montmorillonite [16]. Besides, the protonation of amine groups can allow active monomers to exchange cations on a montmorillonite surface for charge balance [37].

### 3.3. Change in the Basal Spacing of NIPAM/Bentonite with the Temperature

As discussed above, NIPAM monomers can be significantly adsorbed into bentonite. Figure 4 shows the interlayer swelling of clay crystal as the function of the NIPAM monomers. By adding NIPAM, the basal spacing of the clay interlayer increased from 18.34 Å to 20.18 Å in comparison with curves K_b_(30) and K_N_(30), and the expansion was approximately 2 Å. Therefore, the inserted NIPAM monomer may tend to lay flat on the clay surface and cannot remove the adsorption water from clay interlayer without sufficiently many carbon chains [16,37]. 

As the bentonite was heated to 50 °C, the basal spacing decreased 1.22 Å in comparison with curves K_b_(30) and K_b_(50), hence, some of the adsorption water evaporated. However, the basal spacing of the composite still exceeded that of pure bentonite in comparison with curves K_b_(50) and K_N_(50), which demonstrated that conversion of the hydrophilic–hydrophobic property of NIPAM/bentonite was still not realized at this temperature. Especially, the basal spacing of the composite was smaller than that of bentonite when the temperature was increased to 70 °C. Hence, adsorbed NIPAM monomers began to displace the water in the clay interlayer because of the cross-linking of some adsorbed NIPAM monomers on the adsorption site of bentonite [38], which resulted in an increase in the number of hydrophobic alkyl groups [39,40]. However, the basal spacing of the composite was close to that of the pure bentonite at the higher temperature (100 °C) in comparison with curves K_b_(100) and K_N_(100), which indicated some of NIPAM monomers had evaporated.

### 3.4. The Excitation of NIPAM/Bentonite in KPS Aqueous Solution

Figure 5a describes the switching of NIPAM/bentonite in KPS aqueous solution. Thermoresponse of NIPAM/bentonite in KPS aqueous solution was investigated via the change of transmission, as shown in Figure 5b. First, the transmission of the composite decreased gradually with the increase in room temperature (20 °C) to 60 °C. Then, the transmission decreased significantly to the minimum at 70 °C, which indicated the generation of NIPAM polymers and the beginning of thermoresponse of NIPAM monomers.

The FTIR characterization of the NIPAM/bentonite after excitation is shown in Figure 5c. In comparison with the characterization of NIPAM/bentonite before excitation, the peak area of C=O band of the acylamide increased and the peak of the C=C of the alkyl group disappeared in of the composite at 70 °C. The peaks at 2850 cm^−1^, 1390 cm^−1^, and 2920 cm^−1^ respectively correspond to the wagging vibration, symmetric stretching vibration, and antisymmetric stretching vibration of –CH_2_ group [41,42,43]. These peaks indicated the polymerization of NIPAM monomers. Besides, the characteristic peak of hydroxyl groups on the surface of bentonite disappeared in the characterization of PNIPAM/bentonite after polymerization, which indicates that produced NIPAM polymers reacted with bentonite. Furthermore, we attempted to investigate the changing of the basal spacing in the clay interlayer (Figure 5d). As shown in Figure 5d, the characteristic peak at 3680 cm^−1^ of montmorillonite disappeared in the composite after excitation, which indicates montmorillonite layers were completely separated [44,45,46] with the radical polymerization of NIPAM monomers. 

Especially, the NIPAM polymers (PNIPAM) grafted on bentonite were easy to cross-link with each other because the experimental temperature was above the low critical solution temperature (LCST) of PNIPAM [47,48]. The measured average particle size increased significantly from 26.85 to 39.42 μm in comparison with Figure 5e,f. Besides, the excited NIPAM/bentonite was stable during experimental temperature between 50 °C and 90 °C (Figure 5b), unless the experimental temperature was decreased to 40 °C. The measured average particle size decreased significantly to 31.26 μm, as shown in Figure 5g; meanwhile, the measuring transmission of the composite increased by 40~50% (Figure 5b). This temperature indicated the stretching of PNIPAM polymers due to hydrophilic amide groups [47,48]. 

### 3.5. Plugging of NIPAM/Bentonite in the Filtration of Water-Based Drilling Fluids 

The water loss of water-based drilling fluids is crucial to the hydration of boreholes and can cause serious engineering problems (e.g., fracture development, formation leakage, and borehole collapse) [49,50]. Therefore, drilling technologists and engineers created a thin film (which is also called a mud cake) on the surface of each borehole, using the difference in the drilling pressure to reduce the water loss of the water-based drilling fluid. This paper modified the bentonite with NIPAM monomers to further improve the plugging of the mud cake. By using a HTHP filtration meter, the performance of NIPAM/bentonite in the filtration of water-based drilling fluids was evaluated, the results are plotted in Figure 6.

The filtration of basic mud increased with the experimental temperature. The value of *V_F_* of WD1 mud increased with the experimental temperature. Almost all the mud was lost when the temperature exceeded 150 °C. With the addition of common Lv-PAC polymers, the filtrations of the WD drilling fluids obviously decreased. Experimentally, the original filtration of the WD drilling fluid decreased from 63 mL to 19 mL. However, the filtration still significantly increased with the temperature of the drilling fluid, which was due to the thermal degradation of the Lv-PAC polymers networks [51,52].

In addition, the filtration of the basic mud can be reduced by replacing common bentonite with modified bentonite. With the increase in temperature, the *V*_F_ value of the DW drilling fluid also increased in the initial phase. However, the filtration was significantly reduced when the experimental temperature of the DW drilling fluid was close to the alterative zone (70–110 °C) due to the improvement of the plugging of the mud cake, as shown in Figure 7. According to the ESEM imaging results, the initial surface of the mud cake was porous when the temperature was below the alterative zone and the pores were mainly filled with high molecular polymers, which exhibited a similar pattern to Lv-PAC polymers [53]. When the temperature was increased to 70 °C, the pores on the surface of the mud cake were further filled with polymers. According to the ESEM imaging results, the surface of the mud cake was more compact than before due to the coalescence of the NIPAM monomers. In this chemical process, the polymerization initiator (KPS) was adsorbed onto the montmorillonite surface with ion exchange and caused the free radical polymerization of NIPAM monomers, which are bound to montmorillonite via hydrogen, and electrovalent and coordinate bonds [38,39,40].

Using a contact angle meter, the conversion of the surface of the composite can be vividly observed, as shown in Figure 7. The hydrophobicity of the mud cake obviously improved with the increase of the contact angle from 68.84° to 88.28° with increasing temperature. Therefore, the adsorbed water was more easily squeezed out in the case of the hydration of a borehole.

In addition, it was reported that the shale formation accounts for approximately 75% of the drilled strata and 90% of borehole instability problems that occurred in the shale formation [54]. Therefore, this paper further investigates the effects of WD4 and DW4 drilling fluids on the linear expansion of shale. According to Figure 8, the expansion of shale significantly increased from 36.37% to 63.89% with the increase of temperature from 30 °C to 110 °C, which was consistent with the plugging of the WD4 drilling fluid and the degradation of the Lv-PAC polymers. The expansion of shale is reduced by almost half, from 42.34% to 22.28% at 70 °C and reduced to 36.44% at 110 °C as a function of the DW4 drilling fluid. Therefore, the NIPAM/bentonite drilling fluid was beneficial for inhibiting the hydration of the shale at high temperature.

### 3.6. Function of NIPAM/Bentonite on the Rheological Characteristics of the Drilling Fluids

The rheology of a drilling fluid is important for its application in drilling engineering, which is closely related to drilling problems such as carrying cuttings, maintaining borehole stability, and improving the drilling speed of the machinery. Therefore, this paper measured the function of the NIPAM/bentonite composite on the rheology of drilling fluids in room temperature (30 °C) by using a Fann 35A viscosimeter [55,56,57], as shown in Figure 9. The shear rate is determined by the geometric structure of the viscosimeter (the distance between the rotor and the hammer is 1.17 mm), and the reading of the rotary viscometer is proportional to the shear stress (the coefficient of the torsion spring is 3.87 × 10^−5^). Therefore, the shear rate and the shear stress can be calculated via the following equations [17,55,58]:(11)$$\tau =0.511\theta_{N},$$
(12)γ=1.703N.
τ is the shear stress, Pa; γ is the shear rate, s^−1^; *N* is the rotation rate, rpm; and *ϴ_N_* is the reading at rotational speed *N*.

To increase the stability of the drilling fluids, Lv-PAC polymers were added to increase the structural strength of the drilling fluids [18]. The rheology of the WD drilling fluids (Figure 9a) was similar to that of the DW drilling fluids (Figure 9b); both had the characteristics of pseudoplastic fluids and were described by the Bingham and Herschel–Bulkely models [59,60], which are expressed by the following formulas: (13)$$\mathrm{Bingham}\;\mathrm{model}:\;\tau =\tau_{o}+\mu_{p}\gamma ,$$
(14)$$\mathrm{Herschel}–\mathrm{Bulkely}\;\mathrm{model}:\;\tau =\tau_{y}+K\gamma^{n}.$$
$\tau_{o}$ is the yield point in the Bingham model, Pa; $\mu_{p}$ is the plastic viscosity, mPa·s; $\tau_{y}$ is the yield point in the Herschel–Bulkely model, Pa; K is the consistency index of the drilling fluid; and n is the flow behavior index of the drilling fluid.

The experimental results demonstrate that the rheology of the DW drilling fluids was similar to that of the WD drilling fluids, as presented in Table 3. Both satisfied the Bingham model but better fit the Herschel–Bulkely model. According to the rheological parameters, the yield point and the consistency index were slightly increased for NIPAM/bentonite relative to pure bentonite, hence, NIPAM/bentonite improves the structural strength of the drilling fluid. In addition, the flow behavior index of the drilling fluid can be maintained well in comparison with the WD and DW drilling fluids, hence, the NIPAM/bentonite composite can be dispersed well in a water-based fluid, possibly because adsorbed NIPAM monomers can improve the hydrophilicity of bentonite if the experimental temperature is below the polymerization temperature of the NIPAM monomers. 

In addition, this paper measured the viscosity of the drilling fluid after rolling at various temperatures, as shown in Figure 10. The viscosity of the WD drilling fluid was obviously reduced by increasing the temperature from 30 °C to 180 °C. The apparent viscosity of the WD1 drilling fluid was reduced by almost half, from 22.0 mPa·s to 13.2 mPa·s, due to the destruction of the gel structure of the drilling fluid with the thermal degradation of the Lv-PAC polymers networks [51,52].

However, the rheology of the DW drilling fluid was more stable. Initially, the viscosity of the DW drilling fluid decreased in the low temperature range (30~50 °C). As the temperature was gradually increased, the viscosity of the drilling fluid obviously increased in the alterative zone and subsequently decreased to the original viscosity at high temperature (180 °C), which demonstrates the self-recovery behavior of the NIPAM/bentonite composite.

In the alterative temperature zone, an initiator that is adsorbed on the clay surface via ion exchange can induce active NIPAM monomers between the clay layers, which can form polymers on the montmorillonite surface when the experimental temperature is near the polymerization temperature (70 °C). Since the reaction temperature exceeds the lower critical solution temperature (32 °C) of the NIPAM polymers, the wire-ball transition will occur and primary particles with obvious hydrophobicity will be formed when the free radical of the NIPAM polymer increases to a threshold length [61,62]. 

However, a change in the viscosity of the DW4 drilling fluid was not observed. In comparison with the WD4 drilling fluid, the viscosity was more stable between 30 °C and 110 °C, and decreased more slowly—from 23.8 to 22.4 mPa·s as the experimental temperature was increased from 110 °C to 180 °C—because the contribution of the degradation of the Lv-PAC polymers exceeded that of the self-recovery of the NIPAM monomers. Overall, NIPAM/bentonite was helpful for improving the stability of the rheology of drilling fluids at high temperature.

### 3.7. High-Performance Drilling Fluid Based on NIPAM/Bentonite

Based on the results for NIPAM/bentonite at high temperature, this paper further utilized NIPAM/bentonite to prepare high-performance drilling fluids by mixing it with inert plugging materials. The drilling fluid composites are listed in Table 4. 

Two basic muds were gradually added to barite until the density of the mud reached 2.0 g·cm^−3^. Then, the drilling fluid was rolled at 10 °C for 16 h. After that, the performance of the drilling fluid was evaluated, the results are listed in Table 5.

Experimentally, the density of the Bent drilling fluid decreased significantly from 2.0 g·cm^−3^ to 1.6 g·cm^−3^, which corresponded to a decrease in the solid content of the drilling fluid. The viscosity and yield point also obviously decreased, hence, the polymer netting ability in the drilling fluid decreased and the corresponding hanging solid property was weakened. The rheology of the NIPAM/Bent drilling fluid was more stable. After rolling, the density of the drilling fluid was slightly decreased, hence, the polymer netting ability was preserved, and the corresponding hanging solid property was satisfactory. However, the filtration of the NIPAM/bentonite drilling fluid still increased as the drilling fluid was cooled down, which corresponds to the degradation of Lv-PAC and CMC polymers at high temperature, and with mechanical rolling friction [63]. However, the filtration of the rolled NIPAM/Bent drilling fluid obviously decreased in the HTHP experiment as the temperature was increased again due to the cross-linking of the NIPAM monomers that were adsorbed on bentonite particles, which was conducive to the formation and plugging of the mud cake. In addition, Figure 11 shows the functions of the Bent and NIPAM/Bent drilling fluids on the hydration expansion of shale. Under the effect of the Bent drilling fluid, the expansion of shale gradually increased with the temperature. Under the effect of the NIPAM/Bent drilling fluid, the plugging of the NIPAM/Bent drilling fluid on shale improved temporarily in a short temperature range (70–110 °C) as the hydration expansion of shale decreased. Experimentally, the NIPAM/Bent drilling fluid was resistant to high temperature (110 °C) and exhibited satisfactory self-recovery behavior, hence, it has potential for high-temperature stratum drilling. 

## 4. Conclusions

Bentonite was modified with NIPAM monomers to temporarily plug water-based drilling fluids at high temperature. Chemical adsorption was involved in the process of bentonite adsorption of NIPAM monomers due to the function of hydrogen bond and cations exchange, and NIPAM monomers (q_m_ = 2.79 mg/g) were still adsorbed onto bentonite at 90 °C at 0.1 MPa even if the NIPAM monomers tended to evaporate. In comparison with pure bentonite, more adsorption water was squeezed out of NIPAM/bentonite when the temperature was increased to 70 °C, along with the decrease of the basal spacing from 18.34 to 15.71 Å. Therefore, the hydrophobicity of the NIPAM monomers in the clay interlayer can be improved by increasing the temperature, which is because of the increase in the proportion of alkyl chains with the cross-linking of NIPAM monomers.

Based on the NIPAM/bentonite composite, this paper prepared the basic drilling fluid with the initiator KPS and Lv-PAC polymers. For the DW drilling fluids, the alterative zone was in the temperature range between 70 °C and 110 °C, in which a significant improvement in the plugging of water-based drilling fluids was realized, where the filtration can be reduced by approximately 50–70%. With the characterization of the surface of the mud cake, the conversion of the hydrophilic–hydrophobic property and the self-recovery behavior of the mud cake were observed with the changing temperature. As the experimental temperature was increased, a compact mud cake temporarily plugged the water-based drilling fluid at 110 °C, with the contact angle increasing from 68.84° to 88.28° and inhibition of the shale expansion decreasing from 63.89% to 36.44% at 110 °C. 

Based on the research discussed above, this paper further prepared a high-density drilling fluid (ρ = 2.0 g/cm^3^) with NIPAM/bentonite. The rheology of the high-density drilling fluid (NIPAM/Bent) was stable before and after rolling. The NIPAM/Bent drilling fluid exhibited self-recovery behavior at 110 °C and effectively reduced the hydration expansion of shale over the designed temperature range (70–110 °C). Therefore, NIPAM/Bent has high potential for drilling high-temperature stratum with self-recovery behavior.

## Figures and Tables

**Figure 1 materials-12-02115-f001:**
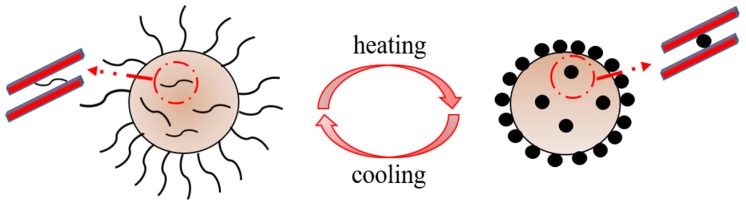
Conversion of the morphology of a N-isopropylacrylamide (NIPAM)/clay composite via temperature changes.

**Figure 2 materials-12-02115-f002:**
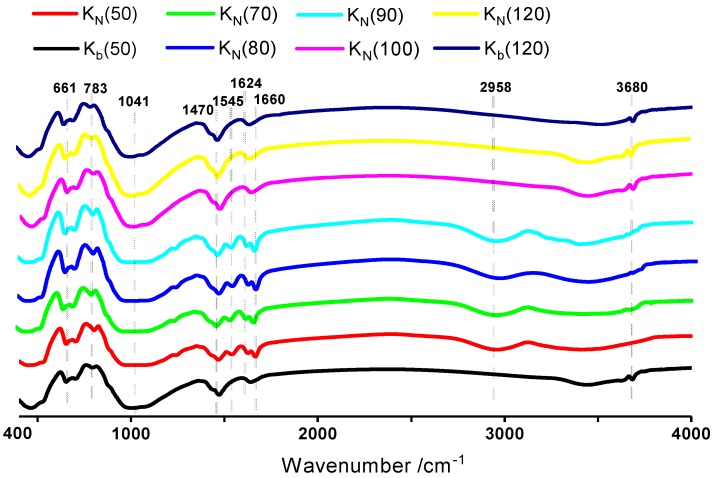
FTIR analysis of NIPAM/bentonite at various temperatures. K_b_(X) refers to the pure bentonite; K_N_(X) refers to the hydrated NIPAM/bentonite; and X refers to the drying temperature, °C.

**Figure 3 materials-12-02115-f003:**
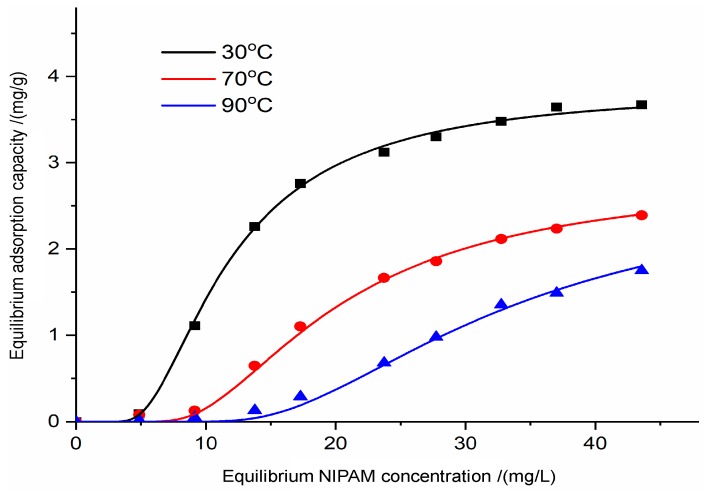
Adsorption isotherms of NIPAM on bentonite.

**Figure 4 materials-12-02115-f004:**
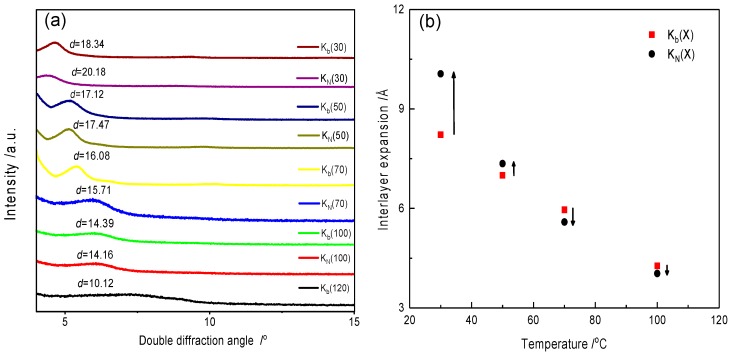
Effect of the adsorbed NIPAM on the swelling of the clay crystal as a function of the temperature. (**a**) Shows basal spacing of the hydrated bentonite and NIPAM/bentonite at different temperatures; and (**b**) describes the difference of interlayer expansion of both samples. K_b_(X) corresponds to the hydrated bentonite; X denotes the drying temperature, °C; K_N_(X) corresponds to the hydrated NIPAM/bentonite; X denotes to the heating temperature, °C; and interlayer expansion represents the expansion of the basal spacing in the clay interlayer, which is based on that of dried clay (120 °C). The up arrow denotes the increase of interlayer expansion with the number of NIPAM monomers; the down arrow denotes the decrease of interlayer expansion with the NIPAM monomers.

**Figure 5 materials-12-02115-f005:**
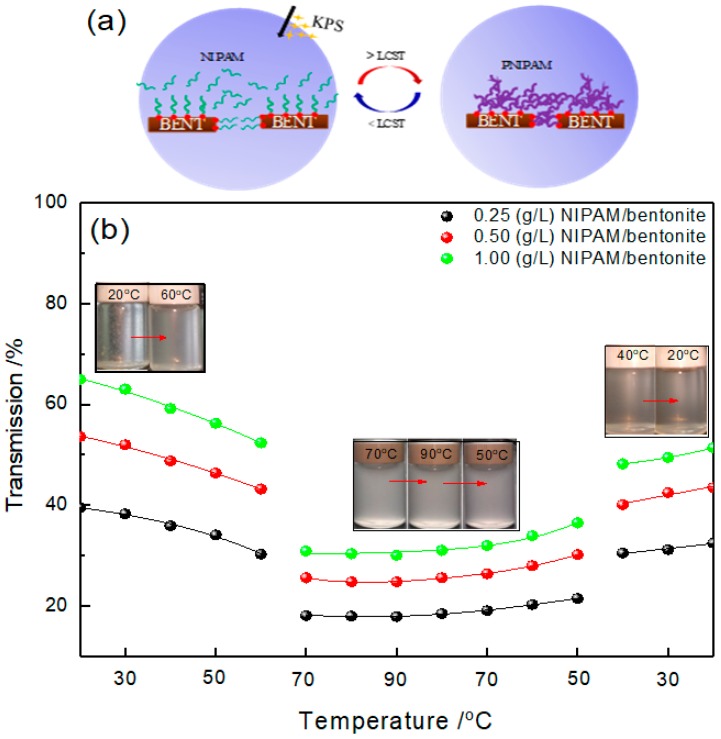
The switching of NIPAM/bentonite in KPS aqueous solution. (**a**) Describes the switching of NIPAM/bentonite in KPS aqueous solution; (**b**) shows the transmittance of NIPAM/bentonite in KPS aqueous solution at various temperatures; (**c**,**d**) respectively show the characteristic peaks of NIPAM/bentonite composite before excitation (at 20 °C) and after excitation (at 70 °C) in FTIR and XRD analysis; (**e**,**f**) respectively describe the particle size of NIPAM/bentonite in KPS aqueous solution at 20 °C and 70 °C; while (**g**) shows the particle size of excited NIPAM/bentonite at 40 °C.

**Figure 6 materials-12-02115-f006:**
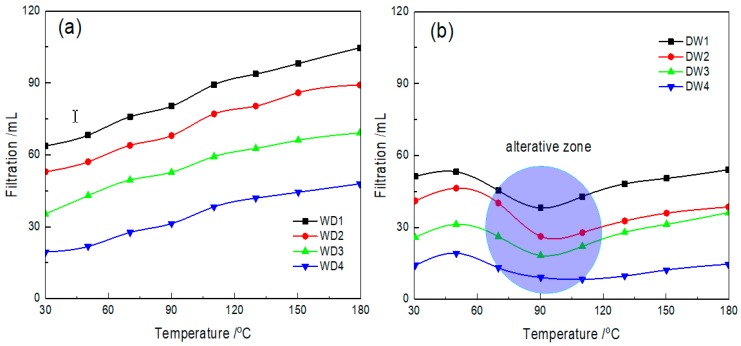
Effects of NIPAM/bentonite on the filtration of the drilling fluid at various temperatures. The experimental pressure was 3.5 MPa; (**a**,**b**) respectively refer to the filtrations of WD drilling fluids and DW drilling fluids at various temperatures.

**Figure 7 materials-12-02115-f007:**
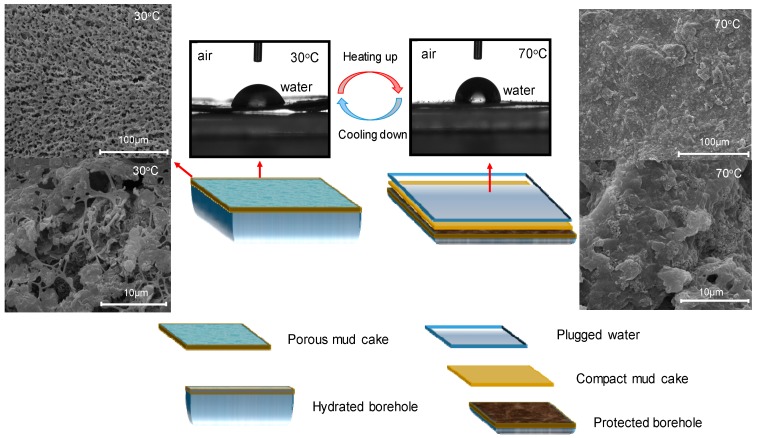
Change in the plugging of the DW4 drilling fluid with temperature.

**Figure 8 materials-12-02115-f008:**
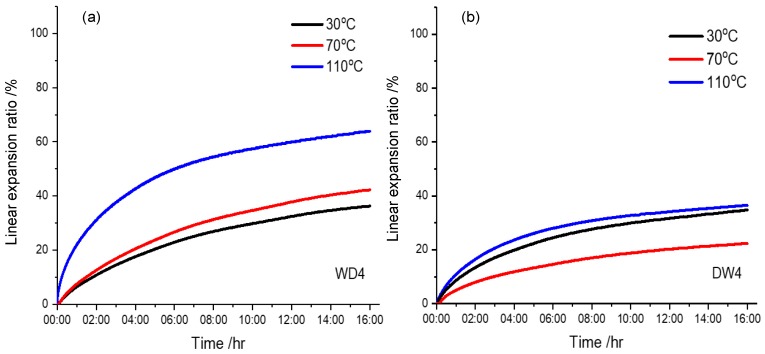
Linear expansion of shale with drilling fluids at various temperatures. The experimental pressure was 3.5 MPa; (**a**,**b**) respectively refer to the functions of WD4 drilling fluids and DW4 drilling fluids on shale expansion at various temperatures.

**Figure 9 materials-12-02115-f009:**
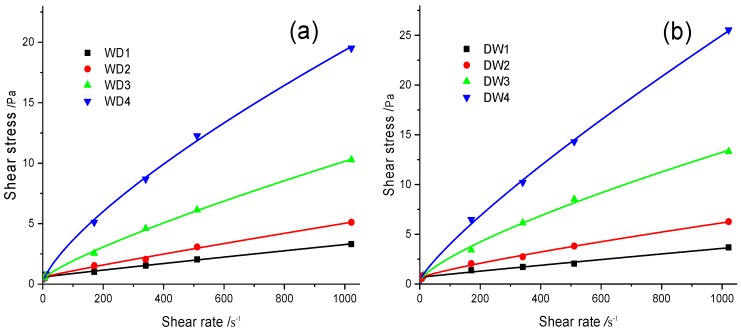
Effects of NIPAM/bentonite on the viscosity of the drilling fluid at various shear rates. (**a**,**b**) respectively show the shear stresses of WD4 drilling fluids and DW4 drilling fluids at room temperature (30 °C).

**Figure 10 materials-12-02115-f010:**
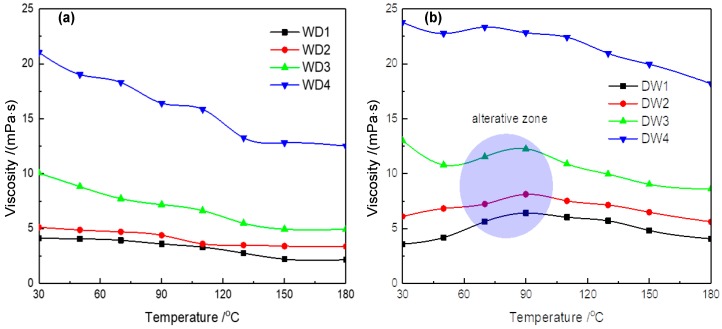
Change in the viscosity of the drilling fluid after rolling at various temperatures. (**a**,**b**) respectively show the viscosities of WD drilling fluids and DW drilling fluids at various temperatures.

**Figure 11 materials-12-02115-f011:**
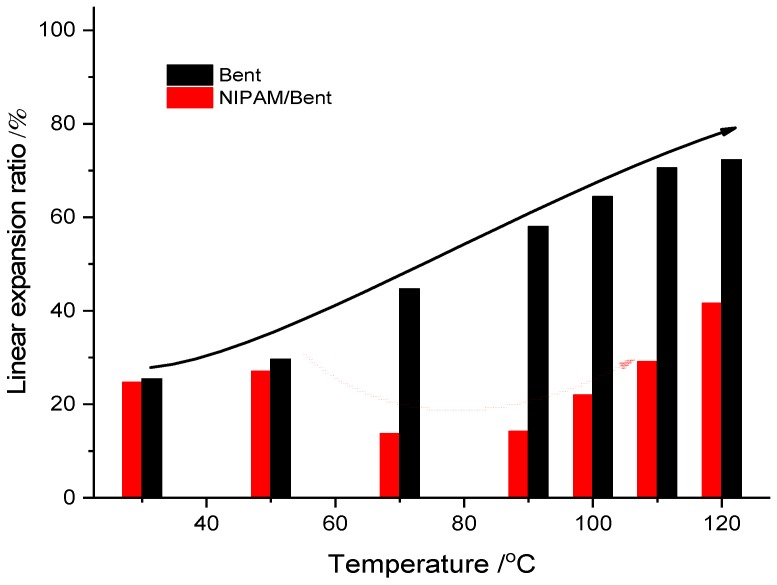
Functions of the Bent and NIPAM/Bent drilling fluids on the shale expansion.

**Table 1 materials-12-02115-t001:** Composites of the water-based drilling fluids.

ID	Composite	Style
WD1	4 wt% bentonite + 0.3 wt% potassium peroxydisulfate (KPS)	Basic mud
WD2	4 wt% bentonite + 0.3 wt% KPS + 0.2 wt% low-viscosity polyanionic cellulose (Lv-PAC)	
WD3	4 wt% bentonite + 0.3 wt% KPS + 0.5 wt% Lv-PAC	
WD4	4 wt% bentonite + 0.3 wt% KPS + 1.0 wt% Lv-PAC	
DW1	4 wt% NIPAM/bentonite + 0.3 wt% KPS	Modified mud
DW2	4 wt% NIPAM/bentonite + 0.3 wt% KPS + 0.2 wt% Lv-PAC	
DW3	4 wt% NIPAM/bentonite + 0.3 wt% KPS + 0.5 wt% Lv-PAC	
DW4	4 wt% NIPAM/bentonite + 0.3 wt% KPS + 1.0 wt% Lv-PAC	

**Table 2 materials-12-02115-t002:** Adsorption isotherm parameters for the adsorption of NIPAM onto bentonite.

Isotherm	Parameters	T/°C		
		30	70	90
Langmuir	q_m_ (mg/g)	3.673	2.679	1.914
	b (L/mg)	0.119	0.053	0.040
	R^2^	0.6949	0.7218	0.6143
Freundlich	n (g/L)	0.620	0.537	0.596
	K_F_ (L/mg)	0.0113	0.0027	0.0034
	R^2^	0.4804	0.7910	0.9741
Temkin	B	1.710	0.496	0.421
	A (L/mg)	0.238	0.799	0.386
	R^2^	0.9764	0.6900	0.6391
D–R	q_m_ (mg/g)	3.850	2.831	2.787
	B_1_ (mol^2^ kJ^−2^)	0.00161	0.00170	0.00177
	E	17.62	17.13	16.82
	R^2^	0.9988	0.9977	0.9929

**Table 3 materials-12-02115-t003:** Rheological parameters of WD and DW drilling fluids at room temperature (30 °C).

Rheology	Parameters	WD Drilling Fluids				DW Drilling Fluids			
		WD1	WD2	WD3	WD4	DW1	DW2	DW3	DW4
Bingham	μ_p_ (Pa⋅s)	0.0027	0.0044	0.0096	0.0188	0.0029	0.0055	0.0126	0.0244
	τ_o_ (Pa)	0.6003	0.6370	0.8358	1.3800	0.6834	0.8198	1.1305	1.2965
	R^2^	0.9952	0.9943	0.9897	0.9748	0.9843	0.9889	0.9794	0.9923
Herschel–Bulkely	τ_y_ (Pa)	0.5738	0.5686	0.3943	0.0811	0.6552	0.5893	0.2944	0.3339
	K (Pa⋅s^n^)	0.0038	0.0076	0.0374	0.1223	0.0041	0.0194	0.0773	0.0809
	n	0.9514	0.9228	0.8055	0.7323	0.9521	0.8188	0.7413	0.8281
	R^2^	0.9957	0.9956	0.9934	0.9990	0.9879	0.9966	0.9974	0.9993

**Table 4 materials-12-02115-t004:** Composites of the water-based drilling fluids.

ID	Composite	ρ (g·cm^−3^)
Bent	4.0 wt% bentonite + 1.0% KPS + 1.0 wt% Lv-PAC + 0.5 wt% CMC + 3 wt% CaCO_3_ + barite	2.0
NIPAM/Bent	4.0 wt% NIPAM/bentonite + 1.0% KPS + 1.0 wt% Lv-PAC + 0.5 wt% CMC + 3 wt% CaCO_3_ + barite	2.0

**Table 5 materials-12-02115-t005:** Performances of high-density drilling fluids before and after rolling.

ID	Rolling	ρ (g·cm^−3^)	μ_a_ (mPa·s)	μ_p_ (mPa·s)	τ_o_ (Pa)	V_F_ (mL)	V_HTHP_ (mL)
Bent	Before	2.0	72	61	11.35	3.5	5.8
	After	1.6	58	50	8.12	5.5	7.2
NIPAM/Bent	Before	2.0	69	57	12.05	3.6	1.2
	After	1.9	65	53	11.86	4.8	2.5

ρ, μ_a_, μ_p_, and τ_o_ refer to the density, apparent viscosity, plastic viscosity, and yield point, respectively; V_F_ refers to the filtration of the drilling fluid at 0.1 MPa at 30 °C; and V_HTHP_ refers to the filtration of the drilling fluid at 3.5 MPa at 110 °C.

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
