# Peer review of "Thermoresponsive Bentonite for Water-Based Drilling Fluids"

_materials, 2019, doi:10.3390/ma12132115_

Round 1
Reviewer 1 Report
In the Manuscript materials-525232 thermo-responsive bentonite for water-based drilling fluids was investigated by authors: Wenxin Dong, Xiaolin Pu, Yanjun Ren.
This paper endowed bentonite with a thermo response via the insertion of N-isopropylacrylamide (NIPAM) monomers. The topic of the manuscript is in accordance with the topic of the journal. In general paper contains a lot of results and I think that the number of results is quite enough for publishing. However, the discussion in the manuscript is not good. Results are listed, but Figures are not denoted in a good manner and thus it was very problematic to find relation between description and Figure, diagram, curve…Also, some of the results are cursory explained. Everything is very confusing. Also, in the description of the results the comparison of the presented results with those found in the literature is missing, as well as using reference(s) for claims and explanations. For that reasons, I suggest the authors to withdraw the paper and rearrange it in a way so that anyone who reads it may understand, the results must have traceability and connectivity, and claims must be reinforced with references. Only such a paper can be published and in accordance with the rules of the journal and contribute to increasing the reputation of the journal itself. For that reason, I suggest the Editor to not accept the paper for publishing.
Some of my comments are listed below:
In experimental part in line 86 the authors did not show amount of the bentonite and the volume in which it was added.
In Line 87 authors did not show the conditions under which UV-VIS measuring was performed. What was measured, under which procedure? More details are required.
Text from 2.5 should be moved to line 86.
In line 122 at which temperatures?
At Figure 2, denote peaks. Also FTIR analysis is not the best choice for measuring and determining chemical composition of the sample. It is much better if classical chemical analysis was used.
In 3.1 references are needed. Spectral band in range 400-1000 cm-1 are not explained. All 3.1 was not explained in a satisfactory level and must be rewritten in a much better way.
In 3.2. what are potential places for chemisorption? References are missing. Results were not explained on a satisfactory level, and only are given cursory explanation. How the S-shape of the curves in Figure 3 may be explained?
In 3.3 references are needed for comparison and explanation of the results.
In 3.4. Figure 5 is confusing. Explanations are confusing. References are missing.
In 3.5 at Figure 6 it is not denoted what is a) what is b). The scale at Figure 6b must be changed, the curve is out of the range.
In 3.6 it is not denoted on best way for which figure and curve is discussion. Comparison with the literature is missing.
Final comment is quantity of the paper is satisfactory, but quality of the paper is not good. More dipper analysis is required.
Best regards
Reviewer 2 Report
The reviewer comments of the paper"Thermo-responsive bentonite for water-based drilling fluids”
- Reviewer
The authors presented an article about the thermo-responsive bentonite for water-based drilling fluids. In general, the article is well written and deserves attention. However, there are several points in the article that require further explanation.
Comment 1:
Abstract as a whole sounds good. Show the practical importance and novelty.
Comment 2:
Introduction is well written and understandable. At the end of the introduction, you must list the sections of the article, and briefly describe what is done in each section.
Comment 3:
Formulas in the original article? If not, please quote from the source.
Comment 4:
Link to fig. 3 must be shown earlier figure 3 – Line 183.
Comment 5:
Sign the axle names in Fig. 3, 4, 6, 8, 9, 10, 11.
Comment 6:
In line 342 N is the rotation rate, r/min. Check whether r/min needs to be replaced with rpm.
Comment 7:
It will be useful to add a section of Nomenclature in which to sign all the physical quantities encountered in the article. There are many physical quantities in the text and such a section will help to find the description of the necessary element.
For example,
µp : The plastic viscosity
etc.
Comment 8:
The conclusions in the article are generally good. But more clearly show the novelty of the article and the advantages of the proposed method.
In general, the topic of the article are relevant. The article is interesting and useful, but needs to be improved. I recommend this article for publication in journal "Materials" after minor changes.
Round 2
Reviewer 1 Report
The paper looks better in comparison with the previous version. Thus, I suggest to the Editor to accept the paper for publication. Best regards